# Fast Mitochondria Detection for Connectomics

**Vincent Casser**                                        casser@alumni.harvard.edu
**Kai Kang**                                                   kk3292@columbia.edu
**Hanspeter Pfister**                                        pfister@seas.harvard.edu
**Daniel Haehn**                                             daniel.haehn@umb.edu

## Abstract

High-resolution connectomics data allows for the identification of dysfunctional mitochondria which are linked to a variety of diseases such as autism or bipolar. However, manual analysis is not feasible since datasets can be petabytes in size. We present a fully automatic mitochondria detector based on a modified U-Net architecture that yields high accuracy and fast processing times. We evaluate our method on multiple real-world connectomics datasets, including an improved version of the EPFL mitochondria benchmark. Our results show an Jaccard index of up to 0.90 with inference times lower than 16ms for a $512 \times 512$px image tile. This speed is faster than the acquisition speed of modern electron microscopes, enabling mitochondria detection in real-time. Our detector ranks first for real-time detection when compared to previous works and data, results, and code are openly available.

**Keywords:** Mitochondria Detection, Connectomics, Electron Microscopy, Biomedical Imaging, Image Segmentation

## 1. Introduction

Connectomics research produces high-resolution electron microscopy (EM) images that allow to identify intracellular mitochondria (Schalek et al., 2016; Suissa-Peleg et al., 2016). Limitations or dysfunction within these structures are associated with several neurological disorders such as *autism* and a variety of other systemic diseases such as *myopathy* and *diabetes* (Zeviani and Di Donato, 2004). Studies also suggest that mitochondria may occupy twice as much volume in inhibitory dendrites than in excitatory dendrites and axons (Kasthuri et al., 2015). Identifying mitochrondria is therefore an important task for neurobiological research and requires a fast automatic detection method that keeps up with the acquisition speed of modern electron microscopes. In EM images, mitochondria mostly appear sparsely as dark round ellipses or, rarely, irregular structures with sometimes visible inner lamellae. Despite the relatively high contrast of their membranes, automatically identifying mitochondria is hard since they float within the cells and exhibit high shape variance, especially if the structures are not sectioned orthogonal when the brain tissue is prepared.

A *de facto* standard benchmark dataset for mitochondria detection was published by Lucchi et al. as the *EPFL Hippocampus dataset* (Lucchi et al., 2012b), and is used by segmentation methods based on traditional computer vision methods (Vitaladevuni et al., 2008; Narasimha et al., 2009; Seyedhosseini et al., 2013) and deep neural networks (Cheng and Varshney, 2017; Oztel et al., 2017; Xiao et al., 2018; Urakubo et al., 2019; Mekuč et al.,

2020). While Lucchi's dataset includes a representative selection of mitochondria in large connectomics datasets, the community observed boundary inconsistencies and several false classifications in the accompanying ground truth labelings (Cheng and Varshney, 2017). We introduce the updated version of this benchmark dataset, *Lucchi++*, re-annotated by three neuroscience and biology experts. This dataset is based on Lucchi's original image data and includes as ground truth consistent mitochondria boundaries and corrections of misclassifications. Another mitochondria dataset was released by Kasthuri et al. (Kasthuri et al., 2015). To counter similar boundary inconsistencies as in Lucchi's dataset, our experts also re-annotate these mitochondria segmentation masks in order to provide a second benchmark dataset, *Kasthuri++*. Figure 1 illustrates the annotation refinements.

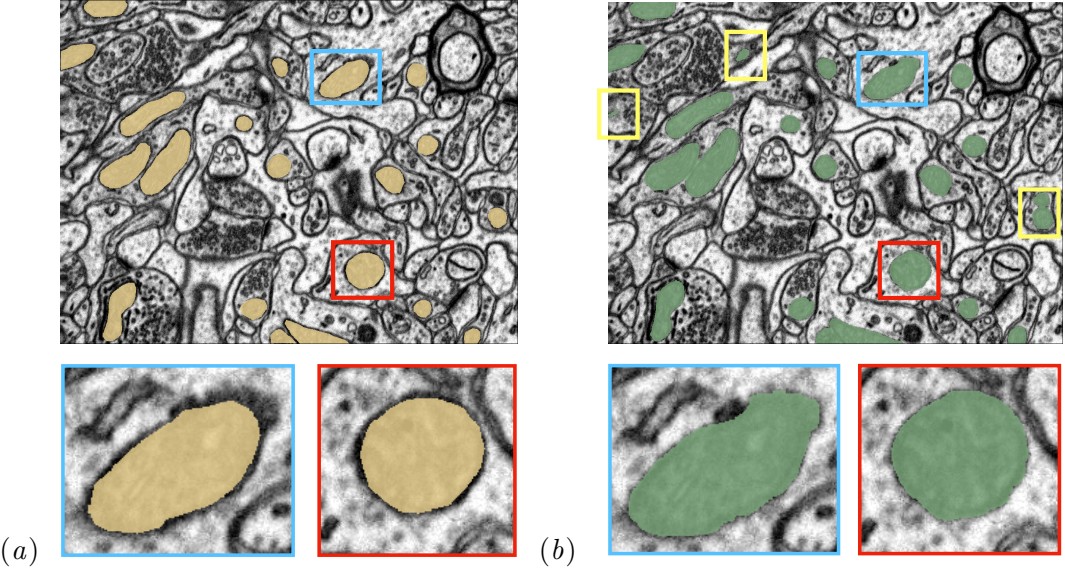

Figure 1: **Our Lucchi++ Mitochondria Benchmark Dataset.** (a) Lucchi et al.'s original EPFL Hippocampus mitochondria segmentation dataset (Lucchi et al., 2012b). (b) Our refined annotation of the dataset to counter boundary inconsistencies (examples in blue and red) and to correct misclassifications (yellow).

With these two datasets, we are able to further study the problem of fast mitochondria detection. We propose an end-to-end mitochondria detection method based on a deep convolutional neural network (CNN). Our method is inspired by the original U-Net approach (Ronneberger et al., 2015), operates purely on 2D images to allow detection without computationally expensive pre-alignment, and is specifically designed to operate at a faster processing speed than the acquisition speed of modern single-beam scanning electron microscopes (11 Megapixels/s) (Schalek et al., 2016). We evaluate our method on Lucchi's original EPFL Hippocampus dataset, the re-annotated Lucchi++ dataset, and the Kasthuri++ neocortex dataset. Our results confirm segmentation accuracy with an IoU (intersection-over-union) within the range of 0.845–0.90 and an average inference speed of 16 milliseconds which is suitable for real-time processing. We compare these numbers to previously published results and rank first among all real-time capable methods, and third

Table 1: **Expert Corrections of Mitochondria Datasets.** We observed membrane inconsistencies and misclassifications in two publicly available datasets. We asked experts to correct these shortcomings in a consensus driven process and report the resulting changes. Experts spent 32-36 hours annotating each dataset.

| Lucchi++ | | | | Kasthuri++ | | |
|---|---|---|---|---|---|---|
| | Before | After | | | Before | After |
| # Mitochondria | 99 | 80 | | # Mitochondria | 242 | 208 |
| Avg. 2D Area [px] | 2,761.69 | 3,319.36 | | Avg. 2D Area [px] | 2,568.38 | 2,640.76 |
| Avg. Boundary Distance | | 2.92 (±1.93) px | | Avg. Boundary Distance | | 0.6 (± 1.36) px |

overall. The created datasets and our mitochondria detection code are available as free and open source software[1].

## 2. Datasets

**EPFL Hippocampus Data.** This dataset was introduced by Lucchi et al. (Lucchi et al., 2012b). The images were acquired using focused ion beam scanning electron microscopy (FIB-SEM) and taken from a $5 \times 5 \times 5\mu m$ section of the hippocampus of mouse brain (voxel size $5 \times 5 \times 5$nm). The whole image stack is $2048 \times 1536 \times 1065$vx and manually created mitochondria segmentation masks are available for two neighboring image stacks (each $1024 \times 768 \times 165$vx). These two stacks are commonly used as separate training and testing data to evaluate mitochondria detection algorithms (Vitaladevuni et al., 2008; Narasimha et al., 2009; Seyedhosseini et al., 2013; Cheng and Varshney, 2017; Oztel et al., 2017). However, the community observed boundary inconsistencies in the provided ground truth annotations (Cheng and Varshney, 2017) and, indeed, our neuroscientists confirm that the labelings include misclassifications and inconsistently labeled membranes (Table 1).

**The Lucchi++ Dataset.** Our experts re-annotated the two EPFL Hippocampus stacks to achieve consistency for all mitochondria membrane annotations and to correct any labeling errors. First, a senior biologist manually corrected mitochondria membrane labelings using in-house annotation software. For validation, two neuroscientists were then asked to separately proofread the labelings to judge membrane consistency. We then compared these judgments. In cases of disagreement between the neuroscientists, the biologist corrected the annotations until consensus between them was reached. The biologist annotated very precisely and only a handful of membranes had to be corrected after proofreading. To fix misclassifications, our biologist manually looked at every image slice of the two Hippocampus stacks for missing and wrongly labeled mitochondria. The resulting corrections were then again proofread by two neuroscientists until agreement was reached. In several cases it was only possible to identify structures as partial mitochondria by looking at previous sections in the image stacks. This could be the reason for misclassifications in the original Lucchi dataset (Figure 1).

**The Kasthuri++ Neocortex Dataset.** We use the mitochondria annotations of the 3-cylinder mouse cortex volume of Kasthuri et al. (Kasthuri et al., 2015). The tissue is

---

1. https://sites.google.com/view/connectomics

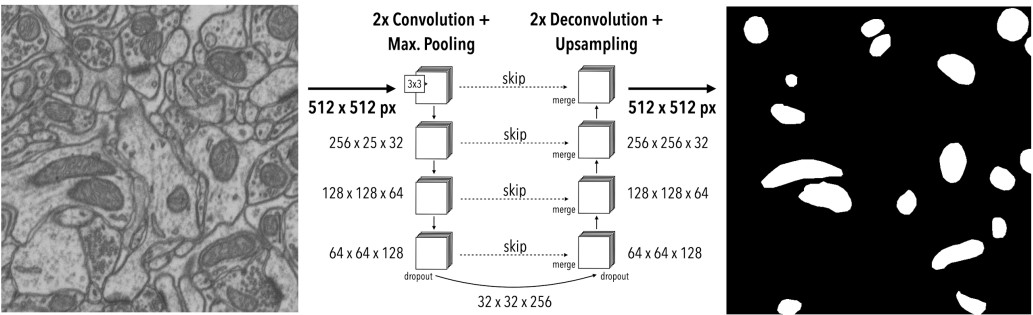

Figure 2: **Our Mitochondria Detector.** We design a light-weight 2D U-Net inspired architecture to output dense predictions at full resolution ($512 \times 512$).

dense mammalian neuropil from layers 4 and 5 of the S1 primary somatosensory cortex, acquired using serial section electron microscopy (ssEM). We asked our experts to correct membrane inconsistencies through re-annotation of two neighboring sub-volumes leveraging the same process described above for the Lucchi++ dataset. The stack dimensions are $1463 \times 1613 \times 85$vx and $1334 \times 1553 \times 75$vx (voxel size $3 \times 3 \times 30$nm).

## 3. Mitochondria Detection

**Architecture.** We build our mitochondria detector by adopting an architecture similar to the 2D U-Net architecture proposed by Ronneberger et al. (Ronneberger et al., 2015). The authors have reported excellent segmentation results on connectomics images similar to ours but target neurons rather than intracellular structures such as mitochondria. Our input and output sizes are $512 \times 512$ pixels, respectively, with the input being fed in as a grayscale image and the output being a binary mask, highlighting mitochondria as the positive class. We exclusively train and predict 2D image slices to allow processing immediately after image acquisition and to avoid waiting for a computationally expensive pre-alignment procedure (Haehn et al., 2017).

**Differences to Original U-Net.** Based on experimental evaluation, we are able to decrease the number of parameters compared to the original U-Net architecture. First, we reduce the number of convolutional filters throughout the network. We then replace transpose convolutions in the decoder with light-weight bilinear upsampling layers that require no parameters. For the encoder, we reach a parameter reduction of 94%, from 19,505,856 to 1,178,480. For the decoder, we reach a 93.6% reduction (from 12,188,545 to 780,053). Lastly, we replace center-cropping from the original U-Net with padding to output densely at full resolution. This modification increases the throughput by an additional 40%. A graphical representation of our architecture can be found in Figure 2.

To verify the effectiveness of our design decisions, we run several ablation studies. We trained both our optimized and the original U-Net using the same setup. On the Lucchi++ dataset, we measure a foreground IoU of 0.888 (overall 0.940) compared to 0.887 (overall 0.939) using the original U-Net, showing equivalent performance. Seemingly the learning capacity of the original U-Net is larger than needed for this task. To investigate, we inspect

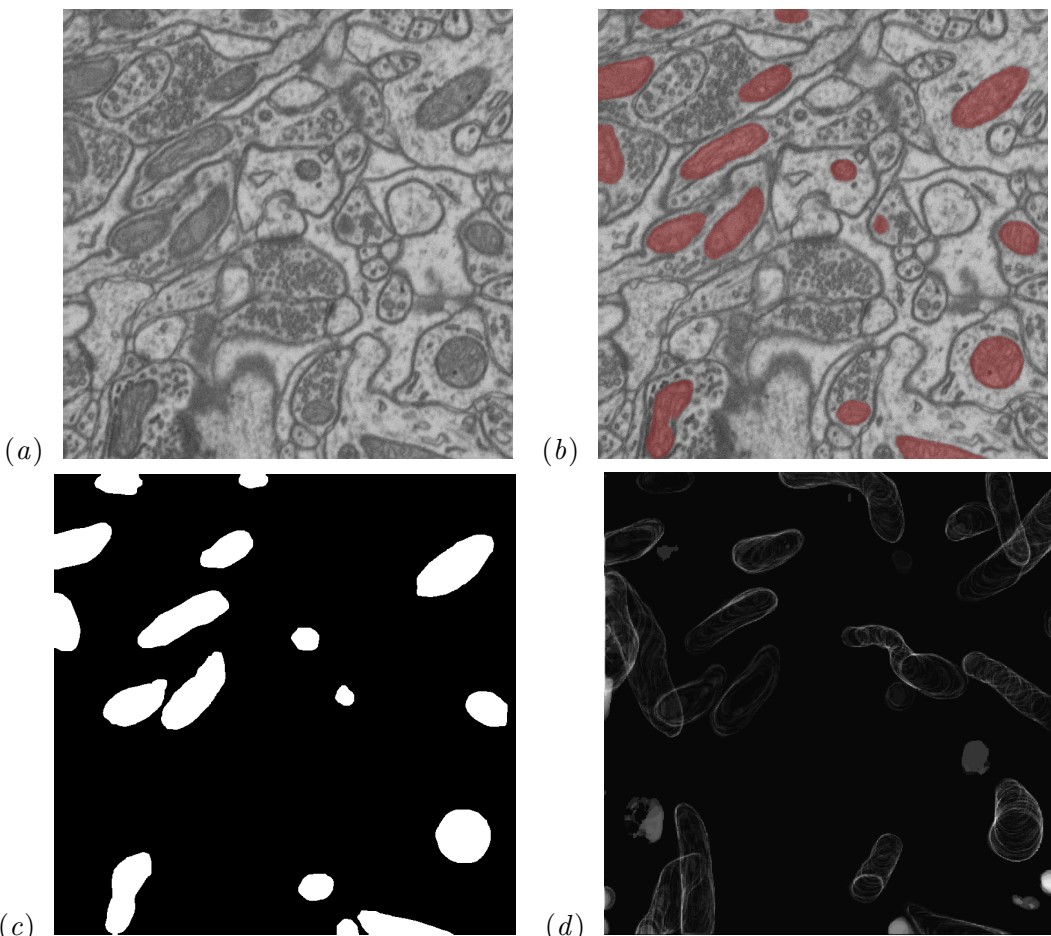

Figure 3: **Example results on EPFL Hippocampus.** (a) An example input image slice. (b) The output of our detector. (c) Expert groundtruth. Our detector finds 16 out of 17 mitochondria in the re-annotated Lucchi++ dataset. (d) The average spatial error distribution of the entire test stack confirms that errors are mainly found at the boundaries.

the trained models for permanently inactive ('dead') ReLU activations and find that in practice, 33.53% of the filters in the original U-Net stay inactive and waste significant compute. In contrast, our optimized U-Net reaches 99.7% utilization.

**Data Augmentation.** We employ an on-the-fly data augmentation pipeline, exploiting known invariances of EM images as much as possible. More specifically, we extract patches in arbitrary orientation and with a varying size that covers at least 60% of the lesser image dimension. This way, each patch covers a large image area and improves robustness towards different voxel sizes. We also apply random bidirectional flipping. Finally, we down- or up-sample the image to $512 \times 512$ before feeding it into the network.

**Training and Inference.** We minimize a standard binary crossentropy loss using Adam optimizer and employ a batchsize of 4 for training. We regularize our network using

a dropout rate of 0.2. Our network fully converges after training for about two hours on a modern Tesla GPU, equivalent to $100,000$ steps. Our detector outputs accurate 2D-segmentation maps. For 3D reconstructions, however, we are able to include additional knowledge across sections as part of post-processing. Inspired by (Oztel et al., 2017), we use a median filter along the $z$-dimension to filter mitochondria which are not present on consecutive sections (*Z-Filtering*).

## 4. Experiments

**Performance Metrics.** Similar to related works, we report segmentation performance as foreground IoU (intersection-over-union/Jaccard-index) and overall IoU. The foreground IoU can be calculated as $J = \frac{TP}{TP+FP+FN}$ where $TP$ are the true positives, $FP$ are the false positives of our positive class (foreground), and $FN$ are false negatives (missing foreground). For a binary classification task like mitochondria detection, the overall IoU is simply defined as the average of the foreground and background IoU. We note that overall IoU is not necessarily a good assessment of classifier performance since the high proportion of background trivially inflates the score, leading to confusion in previous work (Cheng and Varshney, 2017). However, for comparison, we report both measures in addition to inference time and pixel throughput.

**Experimental Setup.** We evaluate our classifier on three datasets. All experiments involve training with the same, fixed parameters (Section 3) and predicting mitochondria on previously unseen test data. Optionally, we apply Z-filtering. We then threshold the predictions and compute similarity measures with respect to manual ground truth labels. For timings, we average multiple runs.

**Datasets.** The EPFL Hippocampus Data is the *de facto* benchmark in the community despite known shortcomings. We evaluate on this dataset and compare against previously reported foreground IoU scores (if not reported, we infer lower bounds for foreground IoU based on the provided overall IoU). We then detect mitochondria in the new Lucchi++ and Kasthuri++ datasets—both with now consistent boundary labelings.

## 5. Results

**Detection Accuracy.** Table 2 summarizes our mitochondria detection performance on previously unseen test images. Averaged across all datasets, we measure a foreground IoU score of 0.870 ($\pm$0.018) in 2D, and 0.879 ($\pm$0.023) with Z-filtering using depth $d \sim 15$. Our average Precision and Recall AUC is 0.979 ($\pm$0.007) and average testing accuracy is greater than 0.993 ($\pm$0.001). We additionally show example qualitative results in Figure 3.

**Inference in Real-time.** The average throughput of our method is between 11 and 35.4 Megapixels/s on a consumer-grade GPU (Table 4). This matches or outperforms the acquisition time of modern single beam electron microscopes (11 Megapixels/s) (Schalek et al., 2016). We are able to process a $512 \times 512$ pixels region on average in 16 milliseconds, allowing mitochondria detection in real-time. We also measure inference speed of Lucchi et al.'s method (Lucchi et al., 2015) with a modern CPU (12x 3.4 GHz Intel Core i7—since it is not executable on GPU)and of the original U-Net (Ronneberger et al., 2015). We also report equivalent timings provided by Xiao et al. (Xiao et al., 2018). Other methods

Table 2: **Mitochondria Detection Results.** We show the performance of our mitochondria detector without and with Z-filtering on three datasets. Z-filtering is beneficial and does not require full 3D stack information. Metrics were computed on the entire 3D volume.

|  |  | Accuracy | Prec. / Recall (AUC) | FG-IoU | IoU |
|---|---|---|---|---|---|
| **EPFL Hippocampus** | 2D U-Net | 0.993 | 0.932 / 0.939 (0.982) | 0.878 | 0.935 |
|  | with Z-filtering | 0.994 | 0.946 / 0.937 (0.986) | 0.890 | 0.942 |
| **Lucchi++** | 2D U-Net | 0.992 | 0.963 / 0.919 (0.986) | 0.888 | 0.940 |
|  | with Z-filtering | 0.993 | 0.974 / 0.922 (0.992) | 0.900 | 0.946 |
| **Kasthuri++** | 2D U-Net | 0.995 | 0.925 / 0.908 (0.969) | 0.845 | 0.920 |
|  | with Z-filtering | 0.995 | 0.932 / 0.902 (0.971) | 0.846 | 0.920 |

Table 3: **Performance Comparison on EPFL Hippocampus.** We compare our method to previous work, ordered by foreground IoU score (the higher, the better). If not reported in the respective papers, we infer lower bounds from the reported overall IoU as indicated by +. More specifically, $IoU_{FG} = 2IoU - IoU_{BG} \geq 2IoU - 1$. Furthermore, $IoU_{BG} \approx 1$ is typically a good approximation given the class imbalance.

Methods inherently requiring pre-alignment are marked as (†).

| Method | Description | FG-IoU | IoU | Real-Time |
|---|---|---|---|---|
| Oztel 2017 (Oztel et al., 2017) | Sliding window CNN + post-proc. | 0.907 | - | |
| Xiao 2018a (Xiao et al., 2018) | 3D U-Net + Res. Blocks | 0.900 | - | |
| Lucchi 2015 (Lucchi et al., 2015) | Working set + inference autostep | 0.895+ | 0.948 | |
| **Ours** | With Z-filtering | **0.890** | **0.942** | ✓ |
| Cheng 2017 (Cheng and Varshney, 2017) | 3D U-Net (†) | 0.889 | 0.942 | ✓ |
| Human Expert | Human trial designed by us | 0.884 | 0.938 | |
| **Ours** | 2D U-Net | **0.878** | **0.935** | ✓ |
| Xiao 2018a (Xiao et al., 2018) | 3D U-Net | 0.874 | - | |
| Cheng 2017 (Cheng and Varshney, 2017) | 2D U-Net | 0.865 | 0.928 | ✓ |
| Cetina 2018 (Cetina et al., 2018) | PIBoost (multi-class boosting) | 0.76 | - | |
| Marquez 2014 (Márquez-Neila et al., 2014) | Random fields | 0.762 | - | |
| Lucchi 2014 (Lucchi et al., 2014) | Ccues + 3-class CRF | 0.741 | - | |
| Lucchi 2013 (Lucchi et al., 2013) | Working set + inference k. | 0.734+ | 0.867 | |
| Lucchi 2012 (Lucchi et al., 2012a) | Kernelized SSVM / CRF | 0.680+ | 0.840 | |
| Lucchi 2011 (Lucchi et al., 2012b) | Learned f. | 0.600+ | 0.800 | |

were not open-source and timings not obtainable. However, based on information about the specific method, we were able to classify whether it can reach real-time performance or not (Table 3).

**Comparison with Other Methods and Human Performance.** We list previously published results on the EPFL Hippocampus dataset in Table 3 and order them by classification performance (high to low). Our detector yields the highest IoU score of all real-time methods. While the difference in accuracy to Chengs' 3D U-Net (Cheng and Varshney, 2017)

Table 4: **Timings.** Our method is able to predict faster than the acquisition speed of modern electron microscopes (11 MP/s) on three datasets. For the EPFL Hippocampus dataset, we compare against Lucchi et al. (Lucchi et al., 2015), Xiao et al. (Xiao et al., 2018), and the original U-Net (Ronneberger et al., 2015)

| | GPU | Full stack [s] | Slice ($512 \times 512$px) [s] | Throughput [MP/s] |
|---|---|---|---|---|
| **EPFL Hippocampus** | | | | |
| Lucchi et al. (Lucchi et al., 2015) | | 815.2 ($\pm$41) | 0.609 ($\pm$0.02700) | 0.16 ($\pm$0.007) |
| Xiao et al. (3D U-Net + Res.) (Xiao et al., 2018) | | 356 | 2.157 | 0.364 |
| Xiao et al. (3D U-Net) (Xiao et al., 2018) | | 265 | 1.552 | 0.490 |
| Ronneberger et al. (2D U-Net) (Ronneberger et al., 2015) | | 22.1 ($\pm$0.824) | 0.030 ($\pm$0.00106) | 5.872 ($\pm$0.227) |
| Ours (2D U-Net) | ✓ | 8.570 ($\pm$0.072) | 0.016 ($\pm$0.00004) | 15.142 ($\pm$0.126) |
| Ours (with Z-filtering) | ✓ | 11.659 ($\pm$0.0082) | 0.023 ($\pm$0.00002) | 11.130 ($\pm$0.008) |
| **Lucchi++** | | | | |
| Ours (2D U-Net) | ✓ | 8.644 ($\pm$0.202) | 0.016 ($\pm$0.00009) | 15.019 ($\pm$0.334) |
| Ours (with Z-filtering) | ✓ | 11.785 ($\pm$0.0141) | 0.022 ($\pm$0.00003) | 11.010 ($\pm$0.013) |
| **Kasthuri++** | | | | |
| Ours (2D U-Net) | ✓ | 4.387 ($\pm$0.0317) | 0.016 ($\pm$0.00006) | 35.421 ($\pm$0.255) |
| Ours (with Z-filtering) | ✓ | 5.122 ($\pm$0.0092) | 0.017 ($\pm$0.00001) | 30.332 ($\pm$0.054) |

is only marginal, we predict single images, require no pre-alignment and thus, even with Z-filtering as post-processing, require less computation for end-to-end processing. Compared to offline methods, we rank fourth with a foreground IoU delta of 0.017. These offline methods are far from real-time capable, with throughput at least 22x lower than ours (Xiao et al., 2018; Lucchi et al., 2015). The best performing method (Oztel et al., 2017) requires extensive sliding-window applications, and multiple CPU-intensive post-processing steps, such as a watershed-based boundary refinement.

We also compare performance to expert annotators on the original EPFL Hippocampus dataset. Some methods, including our 2D U-Net with Z-filtering, yield better IoU scores than human annotators. This is not surprising since CNN architectures are recently able to outperform humans on connectomics segmentation tasks (Lee et al., 2017).

## 6. Conclusions

Our end-to-end mitochondria detector uses 2D images and automatically produces accurate segmentation masks in real-time. This is crucial as connectomics datasets approach petabytes in size. By predicting mitochondria in 2D, processing sections individually can further increase throughput and also support 3D stack alignment with biological priors. We correct the shortcomings and inconsistencies of two existing mitochondria benchmark datasets and provide data and code for free in order to facilitate further research.

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
