# OpenReview forum: "Fast Mitochondria Detection for Connectomics"
_MIDL.io/2020/Conference — MIDL 2020_

### Official Review · AnonReviewer2 · 2020-03-12
**Modified U-Net architecture for mitochrondria detection on enhanced standard datasets**

**Rating:** 4
**Confidence:** 5
**Recommendation:** Oral, Poster

**Summary:**

The authors proposed a modified encoder-decoder (U-net) based architecture for the segmentation of Mitochondria. The proposed architecture has a significantly reduced number of learnable parameters than the original U-Net.  Besides, the authors have re-annotated the standard datasets (Lucchi and Kasthuri) by removing the boundary inconsistencies and incorrect classification. The proposed method is evaluated on these updated datasets and obtained a Jaccard index of 0.90

**Strengths:**

Given the limitations of the existing standard dataset (Lucchi and Kasthuri), there was a need to improve the ground truth fo these datasets. The authors have addressed this issue and re-annotated these datasets. The procedure for re-annotating (and verification) by experts has been very well documented.

Besides, a modified U-net architecture is presented and evaluated on this re-annotated dataset. It is shown that the proposed architecture performs competitively with the existing state of the art methods.

**Weaknesses:**

In my opinion, it would be better to explain in detail the proposed encoder-decoder architecture for segmentation. A detailed figure of the architecture (similar to the one provided in the supplementary materials) would be sufficient. In addition, the following questions need to be answered:

1) It is claimed that reducing the number of learnable parameters from 31 million to 1,958,533 has no impact on precision. This statement should be explained either theoretically or experimentally.
2) In the proposed architecture, the output segmentation map is of the same size as the input map. How is this achieved? By padding?
3) There isn't any 1x1 bottleneck layer in the proposed architecture. Any particular reason for eliminating this?

**Justification Of Rating:**

In my opinion, one of the major strengths of this work is the re-annotation of the existing dataset, which would further drive the research in detection of mitochondria. Besides, the proposed architecture based on U-net performs competitively with the existing methods and obtains a near real-time performance.

**Paper Type:**

both

**Questions To Address In The Rebuttal:**

In my opinion, it would be better to explain in detail the proposed encoder-decoder architecture for segmentation. A detailed figure of the architecture (similar to the one provided in the supplementary materials) would be sufficient. In addition, the following questions need to be answered:

1) It is claimed that reducing the number of learnable parameters from 31 million to 1,958,533 has no impact on precision. This statement should be explained either theoretically or experimentally.
2) In the proposed architecture, the output segmentation map is of the same size as the input map. How is this achieved? By padding?
3) There isn't any 1x1 bottleneck layer in the proposed architecture. Any particular reason for eliminating this?

Also, it would be better to explain the reason for a pixel accuracy of 0.99 on Kasthuri++ and EPFL datasets but a slight disagreement in the jaccard coefficients. ( Jaccard coeff of 0.87 on EPFL and 0.84 on Kasthuri++)

**Special Issue:**

yes

---

> ### Author Response · Authors · 2020-03-27
> **Response**
>
> Thank you very much for the review! Please find our answers below.
>
> Architecture figure: We will expand Supplemental Figure 4 with details on network layers and parameters, and move it to the manuscript.
>
> Parameter reduction without impact on precision: We performed ablation studies and found that the original U-Net (31+ million parameters) does not perform better than our architecture (~2 million parameters). We trained both networks for 1,000 epochs with identical conditions on the Lucchi++ dataset and evaluated without Z-filtering. Our network reaches a Jaccard index of 0.888 and a VOC score of 0.940. The original U-Net reaches a Jaccard index of 0.887 and a VOC score of 0.939.
> Seemingly the learning capacity of the original U-Net is larger than needed for mitochondria detection. Based on this, we also inspect the networks for dead filters that can commonly occur when using ReLU activations. These filters are essentially wasting compute without operating discriminatively on any input. We find that 2,296 of the 6,848 feature maps in the original U-Net are inactive during inference on test samples. Especially 47.2% of filters in the four transposed convolutional layers are inactive. In our network, we replace these layers with light-weight bilinear upsampling layers without parameters. This adds to a total average utilization of 99.7% of filters in our network.
>
> Segmentation output size: Yes, we apply padding to match input and output sizes.
>
> 1x1 bottleneck layer: As in the original U-Net, we do not use intermediate 1x1 bottleneck layers. We use only a single 1x1 output layer to reduce the number of feature maps to the desired number of classes. Our goal is to reduce the computational complexity of the architecture. We did not add additional feature map pooling layers because we apply few (and small) convolutional filters that produce a relatively small number of feature maps.
>
> Pixel accuracy versus Jaccard coefficients: Per-pixel accuracy can be misleading because of the class imbalance between foreground (mitochondria) and background. We still report pixel accuracy to allow comparisons with previous work. The Jaccard score reflects the actual detection performance better, showing that our method performs better on the EPFL data than on Kasthuri++.

---

### Official Review · AnonReviewer3 · 2020-03-13
**Good results / valuable contributions in terms of data and code / lack of novelty**

**Rating:** 3
**Confidence:** 4
**Recommendation:** Poster

**Summary:**

The authors propose a method to segment mitochondria in 3D electron microscopy images. They suggest using a 2D U-net with a reduced number of filters in each convolutional layer to process each of the slices in the 3D volume and hence, speed up the processing time. Additionally, they avoid image alignment between the slices and thanks to this, they achieve a real-time segmentation. While high-resolution connectomics images are usually downsampled, the authors manage to skip this step and even though, get low processing-times.
They validate their approach using three publicly available datasets which have also been used in very similar works.
The code, data and a user interface are freely available but hidden for the review due to the blind policy.

**Strengths:**

In general, the document is well written and the methods are well described, which supports its reproducibility.

The results obtained for both the accuracy of the segmentation and the processing time are quite good.

The authors curate part of the publicly available datasets for connectomics and they provide a new version of it, which is very useful and important for the scientific community.

I would highlight the fact that the entire code and data are freely available as open-source software, which is not always the case. It seems that they also have some annotation framework available (supplementary Figure 1), which if true, can be also very useful.





**Weaknesses:**

The weakest part of this work is that none of the problems to solve nor the proposed approach are innovative. The reduced U-net consists of a network with half the number of filters in each convolutional layer. Besides, the suggested 3D interpolation was already used in Oztel et al., 2017. Indeed the accuracy results of the latter are better, so then where is the improvement in the results reported by the authors? The architecture proposed in Oztel et al., 2017, seems to be much simpler/smaller than the reduced U-Net used here. However, the accuracy measures are better, and the processing time? Also, what can the author say about this? Could it be possible to reduce even more the number of parameters in the U-Net?

Besides, I miss some more recent references such as Chi Xiao et al., Front. Neuroanat. 2018 (https://www.frontiersin.org/articles/10.3389/fnana.2018.00092/full)
where they also train a reduced U-Net with some residual layers and skip connections. Is this approach more accurate but maybe more expensive in terms of memory and time?


In motivation, it is not clear why it is important to get a method faster than the acquisition speed of modern electron microscopes. Could it be possible to integrate this processing into their software? What would be the benefits of it?




**Justification Of Rating:**

In terms of methodology, the work is not novel, but the authors are providing curated datasets and open-source software that will be used by the community in the future.

With little effort during the rebuttal, the work could be a good reference.

**Paper Type:**

validation/application paper

**Questions To Address In The Rebuttal:**

- Table 1, avg. 2D area, what is this measure? is it nm^2 or pixels?

- Correct the links that during the review are anonymous.

- The images used for the training have different voxel sizes. How do you deal with it during the training and the preprocessing? Does the size of mitochondria affect the inference? do you resize the patches that enter in the network? Please give further and explicit details about this in the methodology.

- "Differences to Original U-Net.", please try to be more specific about how the number of parameters was reduced. This information is inferred from the supplementary material but I think it should be stated in the main text.

- In the architecture of the U-Net, in merge-3, the number of channels is 96, why they were not previously reduced to 32 in conv2d-17 to be symmetric?

- During data augmentation, are all the input of the network of the same size? could you please further explain what does this mean?: "we down- or upsample the image before feeding it into the network to map from nanometers to pixels."

- The batch size is not specified. How many patches are cropped from each original image? Or how many patches are used on each epoch? It is said that the training converged after 2 hours. Could you please provide an approximate number of epochs or a measure of convergence?

- Table 2: The results for 2D U-Net were calculated for each 2D slice? or the entire 3D volume and then compared with the Z-filtering?

- "...  compare against previously reported Jaccard scores ... we infer lower bounds through conversion". What do you mean exactly?

- Table 4 is referred to in the text before Table 3.

- From the results in Table 4, can we conclude that, among the cited ones, the only published method with real-time processing is that of Lucchi et al., 2015 ?

- I do not understand what is the aim of this last paragraph: "Proof-of-concept: 3D Alignment with Detected Mitochondria. ". Among the lack of connection with the main test, the authors refer to the mean-square error while this measure was not provided along with the text.

- There is a supplementary material but the authors do not refer to it in the main text.

**Special Issue:**

no

---

> ### Author Response · Authors · 2020-03-27
> **Response**
>
> Thank you for the detailed comments!
>
> Lack of Innovation / Importance: Indeed, we only provide slight modifications of the original U-Net architecture. However, our work targets an important problem in connectomics research: how can we process images fast enough to keep up with the acquisition speeds of modern electron microscopes. We need to avoid computational backlogs since data acquisition of a single volume can involve months of continuous scanning. For this critical application, we need to reach throughputs higher than 11 Megapixels/s. Our modified U-Net architecture reaches this threshold while maintaining high segmentation accuracy. We compare against existing methods, and no other available technique gets even close to our speed. For instance, Oztel et al. provide an architecture that processes 32x32px patches individually. While their base architecture sounds less complicated and yields high detection accuracy, it requires multiple post-processing steps such as outlier detection and watershed-based boundary refinement. This combination is not practical for large connectomics datasets.
>
> Missing references: Thank you for the pointer to the work of Xiao et al. We now include this work and compare performance and timings directly. Their approach indeed is more computationally expensive. The authors report the processing of the EPFL Hippocampus dataset with a maximal throughput of 0.490 Megapixels/s (ours: 11.13 Megapixels/s). Besides, Xiao et al. perform full 3D processing that requires pre-alignment.
>
> Data augmentation and the effect of different voxel sizes, mitochondria sizes, and patch resizing: Our trained U-Net operates on 512x512 image patches. For data augmentation during training, we create many different patches with a randomly varying size that covers at least 60% of the lesser image dimension. This way, each patch covers a large image area and improves robustness towards different voxel sizes. We then up- or down-sample this area to 512x512 pixels. We will expand the manuscript.
>
> Training parameters: We use a batch size of 4 and process around 400 patches per epoch. For each image, our on-the-fly augmentation yields a varying number of patches. On a slightly newer Tesla GPU than our original experimental setup, we now measure 6 seconds per epoch. We trained all networks for 1,000 epochs and did not observe any overfitting. We will include these numbers in the manuscript.
>
> U-Net architecture, parameter reduction, number of channels: We will update the manuscript. First, we reduce the number of convolutional filters throughout the network. We then replace transpose convolutions in the decoder with light-weight bilinear upsampling layers that require no parameters. For the encoder, we reach a parameter reduction of 94%, from 19,505,856 to 1,178,480. For the decoder, we reach 93.6% reduction (from 12,188,545 to 780,053). Second, we replace center-cropping from the original U-Net with padding to output densely at full resolution. This modification increases the throughput by an additional 40%. To demonstrate, we performed ablation studies and will expand the supplemental material (see the response to R2).
> The inputs to the merge-3 layer are asymmetric. The skip-connection layer conv2d-4 (32 filters) provides only low-level information, while the conv2d-17 layer (64 filters) contains more deeply accumulated information. Therefore, we chose not to further reduce the number of filters for conv2d-17 before merging the layers.
>
> Inferred lower bounds on the Jaccard scores: As noticed by Cheng and Varshney 2017, Lucchi reports the Jaccard index as an average of foreground (J_f) and background (J_b), (J_f+J_b) / 2. Other work only includes the foreground Jaccard index (J_f). Since the Jaccard index is bound by 1, we can estimate a lower bound for J_f based on the combined score. A higher value is unlikely since J_b is close to 1 because of the class imbalance. We will update our description at the beginning of section 4 to make this more clear.
>
> Real-time methods / timings in Table 4: Besides Xiao et al., only Lucchi 2015 is publicly available. We specify if a method can reach real-time performance based on published implementation details (Table 3). To allow reproducibility and future comparisons, we release our code and data publicly.
>
> Other comments:
> Table 1, avg. 2D area nm^2 or px?
> We measure the 2D area in pixels and will add [px] to the table.
>
> Table 2, calculated in 2D or 3D?
> We calculated the values directly in 3D. We will update the caption to make this clear.
>
> Table 4 is referred to in the text before Table 3: We fixed the ordering in the manuscript.
>
> Proof-of-concept paragraph feels out-of-place: We will incorporate this paragraph better to present this future research direction.
>
> Supplemental Material: We will refer to the supplemental material in the manuscript when appropriate.
>
> Anonymous links: We will add the links to the final revision. All websites are up and running.

---

> > ### Comment · AnonReviewer3 · 2020-04-03
> > **Reply to authors**
> >
> > I thank the authors for their effort in answering all the questions.
> >
> > - " For instance, Oztel et al. ... it requires multiple post-processing steps such as outlier detection and watershed-based boundary refinement. This combination is not practical for large connectomics datasets."
> > Now your contribution is much clearer. I think you should add a similar explanation in the main text to avoid confusion. Actually, in Table 3, it is only said " Sliding window CNN + post-proc" and even if it is not true, post-processing sometimes seems to be a combination of fast operations.
> >
> > - I would also add the following explanation in the supplementary material: "The inputs to the merge-3 layer are asymmetric. The skip-connection layer conv2d-4 (32 filters) provides only low-level information, while the conv2d-17 layer (64 filters) contains more deeply accumulated information. Therefore, we chose not to further reduce the number of filters for conv2d-17 before merging the layers."
> >
> > I agree with all the remaining comments and I hope they implement all the changes they have mentioned.

---

> > > ### Author Response · Authors · 2020-04-03
> > > **Thank you!**
> > >
> > > We will implement all the changes we mentioned.

---

### Official Review · AnonReviewer4 · 2020-03-19
**Not enough evaluation on speed**

**Rating:** 2
**Confidence:** 4

**Summary:**

It is a validation paper, yet without complete validation. The authors use an existing method on Mitochondria Detection for Connectomics on three datasets including a public dataset. Then they compare the detection performance with those from other literature. They claim the importance of real-time detection, yet did not do a good comparison on speed.



**Strengths:**

The authors perform the proposed method on three datasets, including one public dataset.
They make their dataset and code publicly available.
They collect detection performance from several literature.

**Weaknesses:**

It is a validation paper, which requires , of course, good validation.

The Table 3 shows that the performance of proposed method is worse than comparison methods. But the authors claim that speed is very important for this application, and the proposed method is very quick. This claim is only valid if the authors provide enough validation on speed, which is not provided.

Table 4 is supposed to provide comparison on timing. But the authors only compare the proposed method with one comparison method. If the authors could run other comparison methods by themselves and report time cost, this paper would be acceptable.

**Justification Of Rating:**

The proposed method did not give a very good detection performance. But the authors claim that speed is very important for this application, and the proposed method is very quick. This claim is only valid if the authors provide enough validation on speed, which is not provided.

If the authors could run other comparison methods by themselves and report time cost, this paper would be acceptable.

**Paper Type:**

validation/application paper

**Special Issue:**

no

---

> ### Author Response · Authors · 2020-03-27
> **Response**
>
> We thank the reviewer for valuable comments on this work!
>
> Missing entries in Table 4: Real-time processing requires a method to detect mitochondria faster than the acquisition speed of modern single-beam electron microscopes (11 MP/s). Our method fulfills this criterion. Table 4 so far only included a comparison to Lucchi 2015 since other methods were not publicly available. We reached out to all authors long in advance without success. However, based on published implementation details, we were able to specify if a method can reach real-time performance (Table 3).
>
> We now add two additional timing results from Xiao et al. 2018 (as kindly suggested by Reviewer 3). Their code is not available, but the authors report timings on the EPFL Hippocampus dataset for two methods:
> 3D U-Net: Fullstack 265 seconds, Slice 1.552 seconds, Throughput 0.490 MP/s
> 3D U-Net + Residual Blocks: Fullstack 356 seconds, Slice 2.157 seconds, Throughput 0.364 MP/s
>
> We extended Table 4 with the timings and Table 3 with the Jaccard of 0.900 (3D U-Net + Res.) and 0.874 (3D U-Net). Both methods, tested like ours on Tesla GPUs, are significantly slower than our method (Throughput 11.13+ MP/s).
>
> Our method ranks first among all real-time methods, does not require any pre-alignment (in contrast to Chen 2017 or Xiao 2018), and is openly available.
>
> Other people use our code because it is fast and accurate. For instance, the National Institutes of Health (NIH) uses our method on the Biowulf high-performance computing cluster.
>
> However, we fully understand that the reviewer misses additional direct comparisons with other methods, and we regret that this is not possible. We release our code and our datasets to allow reproducibility and future comparisons.

---

> > ### Author Response · Authors · 2020-03-29
> > **Comparison with Ronneberger's original U-Net**
> >
> > As additional reference, we will include speed measurements of Ronneberger’s original 2D U-Net in Table 4 as shown below (without Z-filtering):
> >
> > 2D U-Net (Ronneberger 2015): Fullstack 22.1 +- 0.824 seconds, Slice 0.0303 +- 0.00106 seconds, Throughput 5.872 MP/s +- 0.227
> >
> > Our optimized U-Net achieves 15.142 Megapixels/s without Z-filtering.

---

### Author Response · Authors · 2020-04-03
**Summary**

Dear Area Chairs and Reviewers,

We would like to thank all reviewers for their valuable comments on this work!

Our work addresses a critical problem in connectomics research: how can we process images fast enough to avoid computational backlogs during the continuous data acquisition with electron microscopes.

Response summary
* AnonReviewer4: we add 3 additional timings and show that our method allows real-time processing almost 3x faster than existing methods
* AnonReviewer3: we answer all questions and clarify the importance of our contribution
* AnonReviewer2: we show that reducing U-Net's complexity from 31 million to 2 million parameters has no impact on precision

Since our method is so fast, the National Institutes of Health (NIH) uses it on the Biowulf high-performance computing cluster.

We think that the open source nature of our mitochondria detector and the two publicly available datasets will be of value to the community.

---

### Meta-Review · Area_Chair1 · 2020-04-07
**MetaReview of Paper327 by AreaChair1**

**Rating:** 3
**Recommendation For Accepted Papers:** Poster

**Metareview:**

Despite some criticism due to limited novelty, the reviewers mostly agree that the authors' work can act as a useful baseline reference for future research in connectomics given that they released new annotations and code for reproducibility. The rebuttal sufficiently addresses the negative review.

**Paper Type:**

validation/application paper

**Special Issue:**

no

---

### Decision · Program_Chairs · 2020-04-11

Accept